# Using Non-Contrast MRA to Discriminate between Obstructive and Nonobstructive Venous Diseases of the Legs

**DOI:** 10.3390/diagnostics11081392

**Published:** 2021-07-31

**Authors:** Chien-Wei Chen, Yuan-Hsi Tseng, Min Yi Wong, Yu-Hui Lin, Teng-Yao Yang, Yin-Chen Hsu, Bor-Shyh Lin, Yao-Kuang Huang

**Affiliations:** 1Department of Diagnostic Radiology, Chia Yi Chang Gung Memorial Hospital and Chang Gung University, Chiayi 61363, Taiwan; chienwei33@gmail.com (C.-W.C.); ymsc10014@gmail.com (Y.-C.H.); 2Division of Thoracic and Cardiovascular Surgery, Chia Yi Chang Gung Memorial Hospital and Chang Gung University, Taoyuan 33323, Taiwan; 8802003@cgmh.org.tw (Y.-H.T.); mynyy001@gmail.com (M.Y.W.); vw200162@gmail.com (Y.-H.L.); 3Department of Cardiology, Chia Yi Chang Gung Memorial Hospital and Chang Gung University, Taoyuan 33323, Taiwan; 2859@adm.cgmh.org.tw; 4Institute of Imaging and Biomedical Photonics, National Yang Ming Chiao Tung University and Department of Medical Research, Chi-Mei Medical Center, Tainan 71004, Taiwan; borshyhlin@gmail.com

**Keywords:** veins, magnetic resonance imaging (MRI), obstructive, not contrast media, MTS

## Abstract

Background: Venous interventions of the legs are less predictable owing to a lock of objective tools. Methods: One hundred and twenty patients with lower extremity venous disease were evaluated anatomically using TRANCE MRI. Then, a QFlow analysis was performed in 53 patients with only one leg affected for hemodynamic evaluation. Those patients with complete QFlow were classified into obstructive and nonobstructive. Results: The QFlow—namely, stroke volume, forward flow volume, mean flux, stroke distance (SD), and mean velocity (MV) in the external iliac vein (EIV), femoral vein (FV), popliteal vein (PV), and great saphenous vein (GSV). The obstructed group had a shorter SD and lower MV in the EIV, EIV/FV, and GSV/PV (SD: *p*-values of 0.025, 0.05, and 0.043, respectively; MV: *p*-values of 0.02, 0.05, and 0.048, respectively). A good performance in discriminating obstructive venous disease was reported for SD in the EIV (area under the curve (AUC) = 67.9%, 95% confidence interval (CI) = 53.2–82.7%), EIV/FV (AUC = 72.4%, 95% CI = 58.2–86.5%), and GSV/PV (AUC = 67.9%, 95% CI = 51.7–84.1%). The SD in the EIV, EIV/FV, and GSV/PV had the ability to discriminate between obstructive and nonobstructive diseases (*p*-values of 0.025, 0.005, and 0.043). The MV in the EIV, EIV/FV, and GSV/PV had ability to discriminate between obstructive and nonobstructive venous diseases (*p*-values of 0.02, 0.005, and 0.048). Conclusions: The SD and MV were lower for obstructive than nonobstructive disease in the EIV.

## 1. Introduction

Venous diseases in the lower extremities range from minor varicose veins to static ulcers and from ambulatory venous hypertension to potentially deadly occlusive diseases, such as deep vein thrombosis (DVT) [1,2,3,4,5,6]. Few options are available for objective venous evaluations of the lower limbs. The venous system is not substantially enhanced by computed tomography (CT) venograms, and high-quality enhancements require specific access from morbid limbs. Compared with conventional angiography, most magnetic resonance venography (MRV) techniques involving contrast media have been proven to be have a favorable sensitivity in the detection of lesions in vessels [7]. However, nephrogenic systemic fibrosis (NSF) is a dangerous complication resulting from the use of gadolinium-based agents. The triggered angiography non-contrast-enhanced (TRANCE) technique records differences in the vascular signal intensity during the cardiac cycle for the subsequent image subtraction and produces vascular images without the use of contrast agents. Our clinical application of this technique has led to innovations in anatomic imaging performances for the whole venous system in the lower extremities [6,8,9]. TRANCE magnetic resonance imaging (MRI) not only demonstrated the locations of venous compressions but also revealed all major collateral veins for improved venous ablation, without the need for venipuncture. We also conducted the QFlow analysis, which can quantitatively measure the hemodynamics of a region of interest. The QFlow technology provides a new noninvasive method for diagnosing hemodynamics by allowing doctors to calculate the flow-related parameters throughout the cardiac cycle. Currently, the QFlow analysis is being used in research related to the cardiovascular system and cerebrospinal fluid [10,11,12]. In this study, we focused on the QFlow analysis and the unique feature of TRANCE MRI to investigate different hemodynamic patterns in patients with venous diseases of the legs.

## 2. Materials and Methods

### 2.1. Patients

This study was approved by the Institutional Review Board of Chang Gung Memorial Hospital (number: 201802137B0). The study involved consecutive patients who were evaluated for venous pathology in their lower extremities using TRANCE MRI at a tertiary hospital between April 2017 and August 2020. We prospectively collected and retrospectively analyzed their data to determine their clinical importance. All patients were suspected of lower extremity venous disease based on the clinical symptom criteria, as follows: leg swelling, itching, pain after standing for long periods, pigmentation around the ankle, leg ulcers, or visible varicose veins. The complete lower-limb venous system in all patients was examined using color Doppler ultrasonography (US) to exclude any emergent statuses (such as acute venous thrombosis). The complete lower-limb venous system (including the femoral vein, great saphenous vein (GSV), popliteal vein, and perforating calf vein) was examined on patients in the supine position. Standard duplex sonography and color Doppler imaging were performed on the same device and reported by two sonographers (YHT and YKH), using defined morphological and hemodynamic criteria. The two sonographers are specialized cardiovascular surgeons and have ten years of experience (YHT) and fifteen years of experience (YKH) in the field of ultrasound examination, respectively. As the pelvic veins were difficult to assess in patients with obesity or edema, these were not evaluated in the regular ultrasonographic survey. The scheduled TRANCE MRI and QFlow analysis were performed within 3 months after the ultrasonography. Patients were excluded if they exhibited poor compliance or had multiple comorbidities that prevented them from lying down for the 1-h TRANCE MRI protocol. Psychological instability or claustrophobia that makes it impossible to cooperate in completing the MRI examination was defined as poor compliance. Initially, 125 patients were enrolled for a TRANCE MRI for venous evaluation of the legs. Five patients were excluded due to pregnancy, non-MRI-compatible ferromagnetic implants, severe arrhythmia, and restless legs. The remaining 120 patients were anatomically evaluated using TRANCE MRI. Segmental QFlow analyses and hemodynamic evaluations were performed for 53 patients where only one leg was involved. Before the scheduled MRI, these patients were divided into obstructive and nonobstructive groups based on the referral reports of the ultrasound examination results (Figure 1).

### 2.2. MRI Acquisition

The MRI was performed using a 1.5 T MRI scanner (Philips Ingenia, Philips Healthcare, Best, the Netherlands), reported by a specialized radiologist (CWC) who has eight years of experience in vascular imaging. The process was performed with patients in the supine position; a peripheral pulse unit trigger was used for imaging. All arterial system images were evaluated using a three-dimensional (3D) turbo spin-echo (TSE) technique during the systole and diastole periods. TSE TRANCE imaging was executed using the following parameters: repetition time (TR), 1 beat; echo time (TE), the shortest; flip angle, 90°; voxel size, 1.7 × 1.7 × 3 mm^3^; and field of view (FOV), 350 × 420 mm^2^. The arterial blood flow is relatively fast during systole, which causes signal dephasing and leads to flow voids. Accordingly, when systolic triggering is applied, the arteries appear black. During diastole, the blood flow in the arteries is slow; the signal is not de-phased, and thus, the arteries appear bright on diastolic scans. Subtracting the two phased scans yields a 3D dataset of the arteries only. Other images of the venous systems were evaluated using 3D TSE short tau inversion recovery (STIR) during systole. TSE STIR TRANCE imaging was executed using the following parameters: TR, 1 beat; TE, 85; inversion recovery delay time, 160; voxel size, 1.7 × 1.7 × 4 mm^3^; and FOV, 360 × 320 mm^2^. STIR provides additional background suppression, because fat and bones are suppressed. When systolic triggering is applied, the arteries appear black. The imaging process yields a 3D dataset of the venous system from which no subtraction is required. A quantitative flow scan is routinely performed to determine the appropriate trigger delay times for systolic and diastolic triggering. All images are acquired without the use of a gadolinium contrast medium [6,9,13,14,15]. In QFlow scans, multiple acquisitions occur within one cardiac cycle, resulting in multiple phases. A QFlow scan on a plane perpendicular to the blood flow was performed to obtain cross-sectional images of the vascular structure of the lower extremity at three levels (Figure 2) [16]. We analyzed the same setpoints for every patient.

A quantitative analysis was performed by drawing the region of interest (ROI) on the vascular lumens (covering the whole lumen) at each set point by a specialized radiologist (CWC), including the external iliac veins, femoral veins, greater saphenous veins, and popliteal veins. Each ROI automatically generated eight variables, including the stroke volume (SV), forward flow volume (FFV), backward flow volume, regurgitant fraction, absolute stroke volume, mean flux (MF), stroke distance (SD), and mean velocity (MV) (Figure 3). In summary, we used TRANCE for morphological imaging and then used a phase contrast GRE technique to measure the blood flow at set points. Both of the external iliac veins, femoral veins, popliteal veins, and greater saphenous veins were analyzed.

### 2.3. Statistical Analysis

Continuous variables (age and QFlow) were analyzed using an unpaired two-tailed Student’s *t*-test or one-way analysis of variance test, and the discrete variables (sex, substance usage, comorbidities, and intervention history) were compared using a two-tailed Fisher’s exact test. The QFlow parameters (including the stroke volume, forward stroke volume, mean flux, stroke distance, and mean velocity in each venous segment) and obstructive venous diseases were evaluated using receiver operating characteristic (ROC) curve analyses. All the statistical analyses were conducted using Data Analysis version 8.0 (Stata Corporation, College Station, TX, USA). The results are presented as the means and standard deviations. The statistical significance was defined as *p* < 0.05.

## 3. Results

Demographic and medical data regarding the 53 patients’ sex, age, substance use, and comorbidities are summarized in Table 1.

The mean age was 59.7 ± 14.1 years, and the majority of patients were female (38, 71.7%). These patients were divided into obstructive and nonobstructive groups based on the referral reports of ultrasound examination results. Patients in the obstructive group were older, had more hypertension, and had more detailed histories of DVT and cancer compared with the nonobstructive group. The morbid legs of the patients with obstructive venous diseases were mostly on the left side.

The TRANCE MRI revealed more cases of May-Thurner Syndrome (MTS) (18:5) and DVT (5:1) in the obstructive group comparing with the nonobstructive group. There were more varicose veins in the nonobstructive group (*n* = 22) compared with the obstructive group (*n* = 14). Varicose veins in the nonobstructive group (17/22) had a trend accompanying prominent great saphenous veins compared with the obstructive group (1/14).

### Discriminant Ability of QFlow According to TRANCE MRI

A QFlow analysis using TRANCE MRI and encompassing the stroke volume (SV, mL), forward flow volume (FFV, mL), mean flux (MF, mL), stroke distance (SD, cm), and mean velocity (MV, cm) was performed for the external iliac veins, femoral veins, popliteal veins, and GSVs in all 53 patients. To minimize the individual bias, we analyzed both the morbid and nonmorbid limbs of each patient. The performance levels of the QFlow parameters (SV, FFV, MV, SD, and MV) for discriminating obstructive venous disease from nonobstructive venous disease were examined against the ratio of morbid to normal limbs for each venous segment (Table 2). A *p*-value < 0.05 was defined as statistically significant. Since five QFlow variables were conducted for multiple comparisons, Bonferroni correction defined the α-value = 0.01 to decrease the type I error in this study. If the *t*-test results reached a *p*-value < 0.05, the area under curve (AUC) was analyzed to define an optimal cutoff value (Table 3).

The SV, FFV, and MF performed poorly (areas under the curve (AUCs) = 50.4–65.3%). Good performance levels in discriminating the obstructive venous disease were reported using the SD in the external iliac vein (EIV) segment (AUC = 67.9%, 95% confidence interval (CI) = 53.2–82.7%), EIV/femoral vein (FV) ratio (AUC = 72.4%, 95% CI = 58.2–86.5%), and GSV/popliteal vein (PV) ratio (AUC = 67.9%, 95% CI = 51.7–84.1%). The MV also demonstrated good discriminative performance in the EIV segment (AUC = 68.7%, 95% CI = 54.2–83.2%), EIV/FV ratio (AUC = 72.6%, 95% CI = 58.6–86.7%), and GSV/PV ratio (AUC = 67.5%, 95% CI = 51.1–83.9%). The SD in the EIV, EIV/FV ratio, and GSV/PV ratio had a discriminative ability between obstructive and nonobstructive venous diseases (*p*-values of 0.025, 0.005, and 0.043; cut-off values of 100.3, 101, and 101.2, respectively) (Figure 4). The MV in the EIV, EIV/FV ratio, and GSV/PV ratio had the ability to discriminate between obstructive and nonobstructive venous diseases (*p*-values of 0.02, 0.005, and 0.048; cut-off values of 100.3, 100.9, and 122.9, respectively).

## 4. Discussion

The venous diseases in the lower extremities include minor varicose veins and static ulcers and range from ambulatory venous hypertension to potentially deadly occlusive diseases, such as DVT. Most patients suspected to have a venous disease of the legs undergo US at the beginning of therapy [17,18]. US is operator-dependent, time-consuming, and inadequate for providing information on the pelvic and abdominal areas. Conventional venography is considered the gold standard for the detection of DVT and other venous occlusive diseases. However, venography is an invasive procedure that requires the use of radiation and contrast agents. Moreover, venography cannot reveal varicose veins beyond the drainage course of the contrast media injection site. CT venography may be useful for the exclusion of a pulmonary embolism in patients with signs of DVT in the legs; however, it still requires the injection of contrast media in the morbid limb to achieve optimal venous imaging of the extremities, which is dangerous for diseased limbs [19]. Multimodal magnetic resonance angiography (MRA) techniques for reconstructing vascular structures mainly include (1) time-of-flight (TOF), (2) phase contrast (PC), and (3) electrocardiographically gated TSE MRA. TOF MRA was applied for evaluating arterial pathologies such as atherosclerosis as early as 1998 [20]. A signal loss in patient vessels resulting from the saturation of the in-plane flow is one of the commonest pitfalls of 2D TOF MRA. Another disadvantage of TOF MRV is that the FOV is small for each image obtained, and it requires considerable time to obtain a complete image of the lower extremities [21,22,23]. MRI with a gadolinium-based contrast media is a relatively rapid method for imaging the lower extremities [24,25]. Although MRI do not involve radiation exposure, the noniodinated contrast agents involved in the imaging process still have undesirable effects. For example, NSF is a dangerous complication of gadolinium-based contrast agents (GBCA) in patients with preexisting impairments of kidney functions and may even occur in patients with normal renal functions, although the recent Canadian guidelines support a more liberal use of the GBCA [26,27]. PC MRI depends on phase shifts caused by the blood flow. Thus, this technique permits the use of coronal or sagittal slice orientations with an FOV along the direction of the vessel of interest and can quantitatively measure the dynamic flow of the chosen region. Researchers have applied PC MRA for evaluating central nervous system pathologies, including vascular disease and hydrocephalus [28,29].

Traditional contrastless MRA, such as TOF MRA and PC MRA, remain time-consuming for imaging the complete vascular structures of the lower extremities. Electrocardiographically gated, multi-step TSE techniques (e.g., TRANCE MRI) offer the possibility of imaging complete vascular structures of the lower extremities in clinical practice. Electrocardiogram gating helps researchers to adapt imaging times to different flow characteristics and, therefore, optimize the image quality more quickly. Although some studies have examined non-contrast-enhanced MRA, most studies have used this technique to evaluate arterial diseases [30,31,32,33,34]. Our study is innovative in that a TRANCE MRI was used investigate the management of complicated lower venous diseases [4,5,6,9]. The examination time may be shortened to less than 30 min by experienced radiological teams (Appendix A). The morphology of the venous anatomy of the lower extremities was clearly imaged in 3D without the use of a contrast media or radiation (Figure 5).

In contrast to CT angiography and among the objective imaging tools, TSE-based subtraction MRI modalities such as TRANCE have unheralded use in hemodynamic discrimination. In this study, we further evaluated the segmental leg vein QFlow data of our previous study. QFlow analyses using TRANCE MRI, encompassing the SV, FFV, ASV, MF, SD, and MV, were performed for the EIVs, FVs, PVs, and GSVs of 53 patients. The MV and SD were found to be more sensitive than the SV, FFV, and MF for differentiating obstructive venous diseases from nonobstructive venous diseases. The QFlow hemodynamic patterns were different between obstructive venous diseases and nonobstructive venous diseases in the SD and MV but not in the SV or MF. The SV, FFV, and MF performed poorly. By contrast, the SD in the EIV segment, EIV/FV ratio, and GSV/PV ratio had effective performances in discriminating obstructive venous diseases. MV also demonstrated a good discriminative performance regarding obstructive venous disease in the EIV segment, EIV/FV ratio, and GSV/PV ratio. The SD in the EIV, EIV/FV ratio, and GSV/PV ratio had the ability to discriminate between obstructive and nonobstructive venous diseases (*p*-values of 0.025, 0.005, and 0.043; cut-off values of 100.3, 101, and 101.2, respectively). The MV in the EIV, EIV/FV ratio, and GSV/PV ratio had a significant ability to discriminate between obstructive and nonobstructive venous diseases (*p*-values of 0.02, 0.005, and 0.048; cut-off values of 100.3, 100.9, and 122.9, respectively). Notably, the nonobstructive venous group had higher GSV/PV ratios in the SD and MV, which may reflect that the majority of the nonobstructive venous leg diseases were attributed to varicose veins in the GSV combined with saphenofemoral junction incompetence.

### Study Limitations

First, there was a possible disagreement between the observers (two sonographers, YHT and YKH) and between the imaging methods (ultrasound and TRANCE MRI), respectively. According to our prior study, the inter-rater reliability between TRANCE MRI and doppler USG showed substantial agreement (Cohen kappa coefficient, 0.73) [15]. These differences between the observers may lead to grouping errors and affect the results of the study. Second, the lack of a reference method of hemodynamic measurements is a big weakness in this study. The major limitations of this study were its nonrandomized design and small sample size. Only patients with typical, unilateral leg-involved venous obstructive diseases were included. However, this was the largest series yet to discuss TRANCE MRIs for venous diseases of the lower extremities. In addition to proving the morphological advantages with minimal radiotoxicity of the TRANCE MRI, this study was the first to analyze QFlow data in clinical scenarios involving venous diseases of the lower extremities.

## 5. Conclusions

In this study, a TRANCE MRI was used to verify obstructions of the venous system in the lower extremities. The QFlow had distinguishing hemodynamic figures. The SD and MV were significantly lower in the EIV for patients with obstructive venous diseases than patients with nonobstructive venous diseases. This promising tool may improve venous intervention strategies in the lower extremities.

## 6. Patents

This project is under the reviewing process at the Taiwan Intellectual Property Office (No 109126307).

## Figures and Tables

**Figure 1 diagnostics-11-01392-f001:**
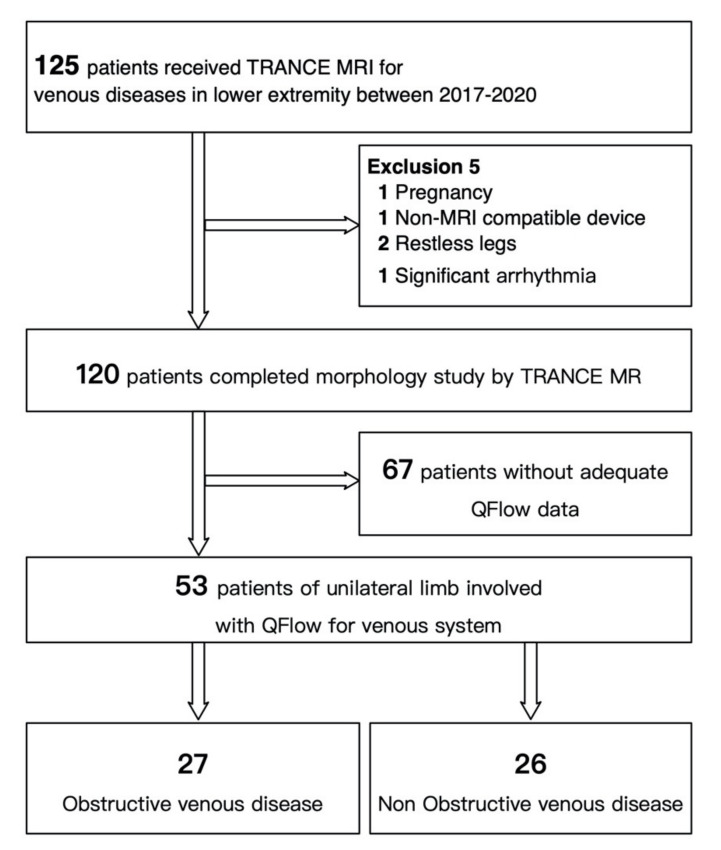
Flow chart of the identification and exclusions of the study cohort.

**Figure 2 diagnostics-11-01392-f002:**
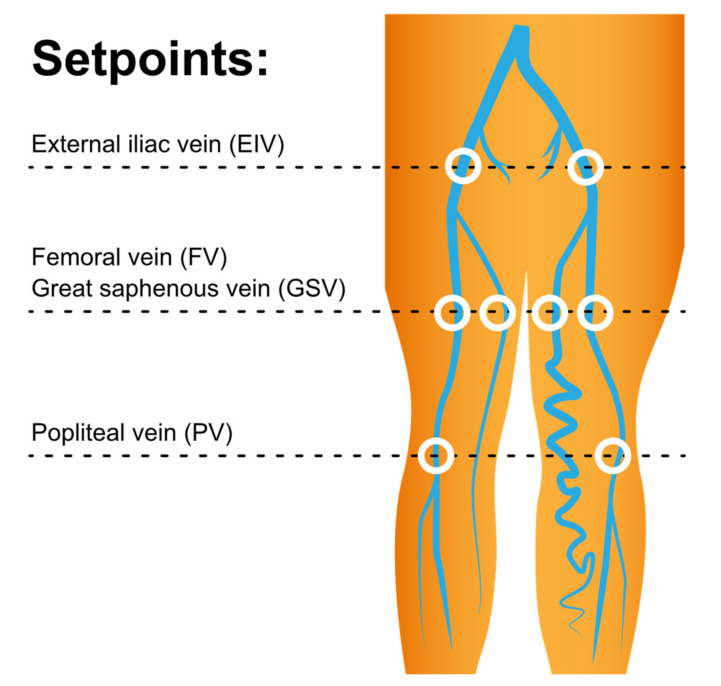
QFlow scan was performed at three levels to obtain the cross-sectional images of the vascular structure of the lower extremity. A postprocessing analysis drew the region of interest on the vascular lumens at the setpoints of both external iliac veins, femoral veins, greater saphenous veins, and popliteal veins. Each setpoint of QFlow analysis can generate eight flow-related variables.

**Figure 3 diagnostics-11-01392-f003:**
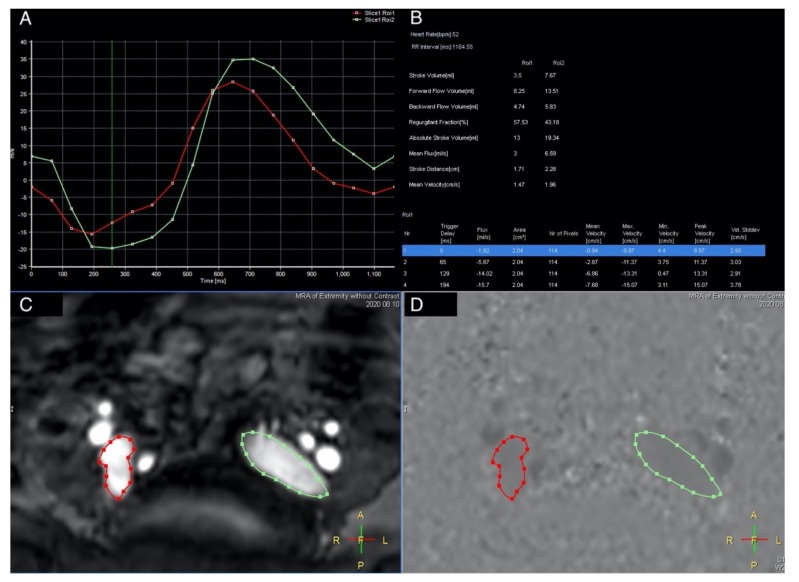
QFlow scan in a TRANCE MRI consisting of the stroke volume, forward flow volume, backward flow volume, regurgitant fraction, absolute volume, mean flux, stroke distance, and mean velocity. (**A**) Two external iliac veins with the flow sequence by time. (**B**) QFlow parameters with different trigger delays. (**C**) Regions of interest of both the external iliac veins. (**D**) Image during data acquisition. (green circle: left external iliac vein, red circle: right external iliac vein).

**Figure 4 diagnostics-11-01392-f004:**
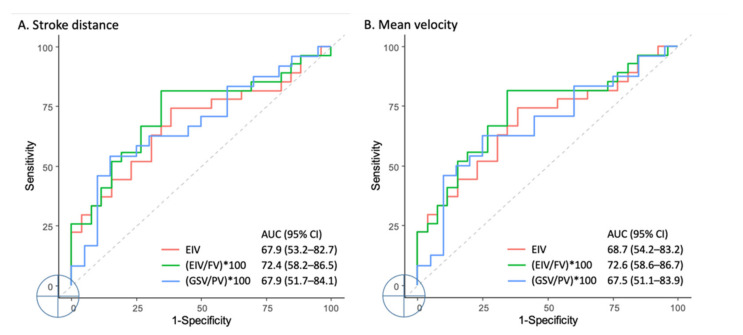
ROC curve of the stroke distance and mean velocity of the QFlow analysis.

**Figure 5 diagnostics-11-01392-f005:**
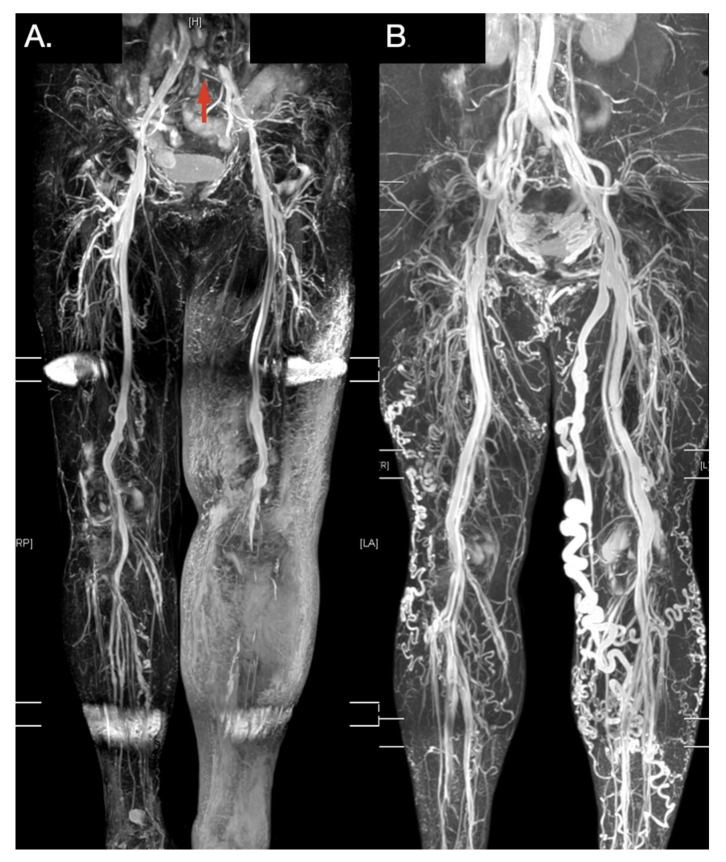
Typical TRANCE MR imaging of obstructive and varicose venous disease. (**A**) Left limb extended subcutaneous edema with a compressed left external iliac vein (Red arrow). (**B**) Lady with left great saphenous varicose vein. Her right thigh received truncal ablation of the right great saphenous vein 4 years ago.

**Table 1 diagnostics-11-01392-t001:** Demographic data of 53 patients of a QFlow analysis by TRANCE MRI.

	Total	Obstructive	Nonobstructive	*p*-Value
Patient number	53	27	26	
Male	15	7	8	0.696
Age (years old)	59.7 ± 14.1	64.7 ± 5.2	54.5 ± 5.4	0.007 *
Substance				
Smoking	10	5	5	0.947
Alcohol	3	2	1	0.575
Betelnuts	3	1	2	0.53
Comorbidities				
Hypertension	15	11	4	0.041 *
Diabetes	8	5	3	0.478
Stroke	0	0	0	NA
Coronary disease	1	0	1	0.322
DVT history	6	6	0	0.011 *
Cancer	8	7	1	0.025 *
Left leg involved	32	23	9	<0.001 *
TRANCE MRI proved varicose veins	36	14	22	0.011 *
Chronic leg ulcer	8	3	5	0.409

DVT: deep venous thrombosis; * *p*-value < 0.05.

**Table 2 diagnostics-11-01392-t002:** The performance of the QFlow parameters (the ratio of morbid limb to normal limb) in discriminating obstructive venous disease.

Variable	Total (*n* = 53)	Obstruction (*n* = 27)	Non-Obstruction (*n* = 26)	*p*-Value
Stroke volume (SV)				
External iliac vein (EIV)	100 (60, 124)	72 (39, 124)	105 (80, 132)	0.059
Femoral vein (FV)	91 (60, 136)	80 (41, 126)	113 (66, 174)	0.084
(EIV/FV) × 100	86 (58, 147)	72 (45, 124)	99 (66, 162)	0.247
Great saphenous vein (GSV)	130 (72, 766)	116 (59, 706)	168 (94, 848)	0.239
Popliteal vein (PV)	89 (53, 120)	92 (50, 114)	78 (60, 207)	0.631
(GSV/PV) × 100	154 (79, 518)	135 (51, 664)	242 (86, 463)	0.423
Forward flow volume (FFV)				
External iliac vein (EIV)	92 (56, 121)	76 (22, 121)	104 (78, 132)	0.072
Femoral vein (FV)	95 (64, 136)	91 (52, 126)	113 (66, 174)	0.135
(EIV/FV) × 100	81 (49, 145)	66 (41, 113)	99 (61, 149)	0.145
Great saphenous vein (GSV)	125 (81, 332)	111 (63, 217)	170 (94, 588)	0.083
Popliteal vein (PV)	92 (63, 120)	97 (63, 114)	78 (60, 207)	0.852
(GSV/PV) × 100	147 (73, 312)	103 (54, 281)	203 (96, 342)	0.157
Mean flux (MF)				
External iliac vein (EIV)	100 (59, 124)	72 (39, 124)	105 (80, 133)	0.070
Femoral vein (FV)	95 (64, 136)	89 (48, 126)	113 (67, 171)	0.126
(EIV/FV) × 100	83 (55, 146)	67 (45, 114)	96 (65, 163)	0.140
Great saphenous vein (GSV)	129 (67, 778)	116 (59, 694)	173 (93, 824)	0.239
Popliteal vein (PV)	88 (61, 120)	92 (52, 114)	77 (61, 209)	0.709
(GSV/PV) × 100	160 (80, 538)	126 (52, 608)	238 (86, 466)	0.370
Stroke distance (SD)				
External iliac vein (EIV)	98 (71, 123)	77 (29, 107)	104 (80, 132)	0.025 *
Femoral vein (FV)	94 (76, 128)	94 (70, 124)	95 (76, 129)	0.594
(EIV/FV) × 100	94 (64, 122)	68 (19, 96)	109 (77, 132)	0.005 *^,b^
Great saphenous vein (GSV)	126 (73, 423)	93 (46, 349)	147 (101, 763)	0.131
Popliteal vein (PV)	91 (57, 119)	97 (61, 130)	79 (54, 118)	0.533
(GSV/PV) × 100	135 (66, 333)	100 (48, 314)	204 (119, 726)	0.043 *
Mean velocity (MV)				
External iliac vein (EIV)	99 (71, 128)	77 (29, 107)	104 (80, 133)	0.020 *
Femoral vein (FV)	94 (76, 128)	94 (70, 123)	95 (76, 129)	0.581
(EIV/FV) × 100	94 (63, 129)	68 (19, 96)	112 (77, 135)	0.005 *^,b^
Great saphenous vein (GSV)	126 (73, 382)	93 (58, 351)	148 (102, 601)	0.172
Popliteal vein (PV)	91 (57, 119)	98 (61, 130)	79 (54, 118)	0.510
(GSV/PV) × 100	135 (66, 332)	100 (49, 316)	206 (119, 502)	0.048 *

Data were presented as the median (25th percentile, 75th percentiles); * *p*-value < 0.05 was defined as statistically significant; ^b^ Bonferroni correction redefined the *p*-value < 0.01 as statistically significant.

**Table 3 diagnostics-11-01392-t003:** The area under curve (AUC) was analyzed to define an optimal cutoff value if the *t*-test results reached a *p*-value < 0.05.

Variable	AUC, % (95% CI) *	Cutoff #	Sensitivity, % (95% CI)	Specificity, % (95% CI)
Stroke distance (SD)				
External iliac vein	67.9 (53.2–82.7) *	≤100.3	74.1 (53.7–88.9)	61.5 (40.6–79.8)
(External iliac vein/Femoral vein) × 100	72.4 (58.2–86.5) *	≤101.0	81.5 (61.9–93.7)	65.4 (44.3–82.8)
(Great saphenous vein/Popliteal vein) × 100	67.9 (51.7–84.1) *	≤110.2	54.2 (32.8–74.4)	85.0 (62.1–96.8)
Mean velocity (MV)				
External iliac vein	68.7 (54.2–83.2) *	≤100.3	74.1 (53.7–88.9)	61.5 (40.6–79.8)
(External iliac vein/Femoral vein) × 100	72.6 (58.6–86.7) *	≤100.9	81.5 (61.9–93.7)	65.4 (44.3–82.8)
(Great saphenous vein/Popliteal vein) × 100	67.5 (51.1–83.9) *	≤122.9	62.5 (40.6–81.2)	75.0 (50.9–91.3)

Data were presented as the median (25th percentile, 75th percentiles); AUC, area under curve; CI, confidence interval; * *p*-value < 0.05; # Determined by the Youden Index.

## Data Availability

The data presented in this study are available on request from the corresponding author. The data are not publicly available due to ethical restrictions.

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
