# Peer review of "Using Non-Contrast MRA to Discriminate between Obstructive and Nonobstructive Venous Diseases of the Legs"

_diagnostics, 2021, doi:10.3390/diagnostics11081392_

Round 1
Reviewer 1 Report
The Authors present the study design entitled “Using Non-Contrast MRA to discriminate between obstructive and non-obstructive venous diseases of the legs.”
This is a very interesting study that raises a significant clinical problem and could be helpful for doctors in their everyday clinical practice.
The proper diagnosis of venous disease is challenging. The Authors aimed to check the usefulness of the new application of MRI with the use of triggered angiography non–contrast-enhanced (TRANCE) methodology and focused on the QFlow analysis as the unique feature of TRANCE MRI, to investigate different hemodynamic patterns in patients with venous diseases of legs.
The reviewer appreciates the Authors’ effort and notices the important clinical value of the study. The study is nicely written. The graphic presentation is out of any comments. The figures are precise and clear. I strongly recommend this study for publication in Diagnostics, but after some comments:
- The information about the numbers of patients included in the study should be in the abstract as well.
- The description of QFlow in the introduction part should be added.
- How the suspicion of having venous pathology in 60 enrolled patients was qualified (clinical symptoms? Objective features?) – which criteria of suspicion were taken into account?
- What was “poor compliance” as an exclusion criterion mean?
Author Response
Reviewer 1
The Authors present the study design entitled “Using Non-Contrast MRA to discriminate between obstructive and non-obstructive venous diseases of the legs.”
This is a very interesting study that raises a significant clinical problem and could be helpful for doctors in their everyday clinical practice.
The proper diagnosis of venous disease is challenging. The Authors aimed to check the usefulness of the new applicationof MRI with the use of triggered angiography non–contrast-enhanced (TRANCE) methodology and focused on the QFlow analysis as the unique feature of TRANCE MRI, to investigate different hemodynamic patterns in patients with venous diseases of legs.
The reviewer appreciates the Authors’ effort and notices the important clinical value of the study. The study is nicelywritten. The graphic presentation is out of any comments. The figures are precise and clear. I strongly recommend thisstudy for publication in Diagnostics, but after some comments:
[Comment 1]
The information about the numbers of patients included in the study should be in the abstract as well.
[Answer 1]
We appreciated deeply for your encourage and warming comments. We have added the information about the numbers ofpatients in the study in the Abstract.
[Change]
[Page 1, abstract line 19]
“Methods: 120 patients with lower extremity venous disease were evaluated anatomically using TRANCE MRI. Then, performed QFlow analysis in 53 patients with only one leg affected for hemodynamic evaluation.”
[Comment 2]
The description of QFlow in the introduction part should be added.
[Answer 2]
We had added the description of QFlow in the Introduction.
[Change]
[Page 2, introduction, line 53-57]
“We also conducted the QFlow analysis, which can quantitatively measure the hemodynamics of a region of interest. QFlow technology provides a new non-invasive method for diagnosing hemodynamics by allowing doctors to calculate flow-related parameters throughout the cardiac cycle. Currently, QFlow analysis has been used in research related to the cardiovascular system and cerebrospinal fluid.”
Three references added for this description.
[Comment 3]
How the suspicion of having venous pathology in 60 enrolled patients was qualified (clinical symptoms? Objective features?) – which criteria of suspicion were taken into account?
[Answers 3]
Those 125 patients were suspected of lower extremity venous disease based on the clinical symptom, criteria as follows: leg swelling, itching, pain after long-standing, pigmentation around the ankle, leg ulcers, or visible varicose veins. All patients underwent ultrasonography, following by the scheduled TRANCE MRI and QFlow analysis. 5 patients were excluded because they meet the contraindications of MRI, another 67 patients were excluded because of lack of adequate QFlow data. Finally, 53 patients of unilateral limb involvement with QFlow data were enrolled for the study. We have added a description of the diagnostic criteria for clinical symptoms of lower extremity venous disease and modified the order of the sentences to make the article more straightforward to be understood.
[Change]
[Page 2, Materials and Method, patients, line 66-73]
“All patients were suspected of lower extremity venous disease based on the clinical symptom, criteria as follows: leg swelling, itching, pain after long-standing, pigmentation around the ankle, leg ulcers, or visible varicose veins. The complete lower-limb venous system in all patients was examined using color Doppler ultrasonography (US) to exclude emergent status (such as acute venous thrombosis). The complete low-er-limb venous system (including femoral vein, great saphenous vein (GSV), popliteal vein, and perforating calf vein) was examined on patients in the supine position. As pelvic veins were difficult to assess in patients with obesity or edema, these were not evaluated in the regular ultrasonographic survey. The scheduled TRANCE MRI and QFlow analysis were performed within 3-months after the ultrasonography.”
[Comment 4]
What was “poor compliance” as an exclusion criterion mean?
[Answer 4]
Psychological instability or claustrophobia makes it impossible to cooperate in completing the MRI examination was defined as poor compliance. We have added the description in the Materials and Methods.
[Change]
[Page 2, line 82-83]
“Psychological instability or claustrophobia makes it impossible to cooperate in completing the MRI examination was defined as poor compliance.”
Thank you for the informative and careful reviewing our manuscript. We learn lot during the revision this article. Please contact me at the following address for additional information.
Sincerely,
Yao-Kuang Huang, MD, PhD
Division of Thoracic and Cardiovascular Surgery
Chia-Yi Chang Gung Memorial Hospital, Putz, Taiwan.
Fax: 886-975368209
E-mail: huang137@icould.com
Reviewer 2 Report
Though the paper is interesting some concern is raised for several reasons. First, it is not clear how the patients were divided into occlusive and non-occlusive group. According to the paper it was done by looking into the referrals. Referrals can be uninformative or misguiding. Who did the selection for the two groups? Was there no doubt when grouping the patients? Was there any second reading of the referrals to ensure that the grouping was correct?
No reference methods are used, which is a big weakness in the study.
You do a lot of p-test from data of the MR exam. It is not clear to me what is measured and where in the venous tree. You state that several setpoints were analyzed – how can I see that in the results? Did you analyze the same setpoints for every patient? You state that SD and MV can separate pts with occlusive from non-occlusive venous disease in EIV. Why only in EIV – do you have an explanation for this finding.
You state that results are ratios between affected and unaffected leg. However, this implies that only patients with unilateral venous leg disease was enrolled? How can a ratio be above 1?
You should do Bonferroni-correction of data when so many tests are performed. This would most likely influence your final results.
How were data collected? You did analyses off-line by one operator? Blinded? Automated? Inter- intraobserver variation?
Table 2 is not very informative. Which are and which aren’t TRANCE findings?
As stated above the results are difficult to read due to many abbreviations in Table 3. Table 3 should be better designed.
Don’t repeat results in the discussion. In discussion is missing the comparison to results of other groups.
When abbreviations are introduced, then spelled out first time. Eg. missing for FV in abstract.
Author Response
[Comment 1]
Though the paper is interesting some concern is raised for several reasons. First, it is not clear how the patients were divided into occlusive and non-occlusive group. According to the paper it was done by looking into the referrals. Referrals can be uninformative or misguiding. Who did the selection for the two groups? Was there no doubt when grouping the patients? Was there any second reading of the referrals to ensure that the grouping was correct?
[Answer 1]
Before the scheduled MRI, these patients were divided into obstructive and nonobstructive groups based on the referral reports of ultrasound examination results. There is indeed possible disagreement between the observers and between the imaging methods, respectively. According to our prior study (Chen CW, et al. Medicine 2021), inter-rater reliability between TRANCE MRI and doppler USG showed substantial agreement (Cohen kappa coefficient, 0.73). These differences between the observers may lead to grouping errors and affect the results of the study Thus, we make a more explicit description of the radiologist (CWC) for reporting MRI and the sonographers (YHT, YKH) and added the associated description and reference.
[Change]
“Before the scheduled MRI, these patients were divided into obstructive and nonobstructive groups based on the referral reports of ultrasound examination results” [page 2 line 88-90]
“These patients were divided into obstructive and nonobstructive groups based on the referral reports of ultrasound examination results” [page 6 line 157-158]
“Standard duplex sonography and color Doppler imaging were performed on the same device and reported by two sonographers (YHT, YKH), using defined morphological and hemodynamic criteria. The two sonographers are specialized cardiovascular surgeons and have ten years of experience (YHT) and fifteen years of experience (YKH) in the field of ultrasound examination, respectively.” [page 2, line 73-76]
“MRI was performed using a 1.5 T MRI scanner (Philips Ingenia, Philips Healthcare, Best, the Netherlands), reported by a specialized radiologist (CWC) who has eight years of experience in vascular imaging.” [page 3, line 95-96]
“There is a possible disagreement between the observers (two sonographers, YHT and YKH) and between the imaging methods (ultrasound and TRANCE MRI), respectively. According to our prior study, inter-rater reliability between TRANCE MRI and doppler USG showed substantial agreement (Cohen kappa coefficient, 0.73). These differences between the observers may lead to grouping errors and affect the results of the study.” [page 10, 277-279]
[Comment 2]
No reference methods are used, which is a big weakness in the study.
[Answer 2]
Thank you for pointing out this weakness of our study. We have added this description in the Study Limitation.
[Change]
Study limitation
“Second, the lack of a reference method of hemodynamic measurement is a big weakness in this study.” [page 10, line 282-283]
[Comment 3]
You do a lot of p-test from data of the MR exam. It is not clear to me what is measured and where in the venous tree. You state that several setpoints were analyzed – how can I see that in the results? Did you analyze the same setpoints for every patient?
[Answer 3] We analyzed the same setpoints for every patient. We have modified the description and added a new figure to clearly display the settings in the venous tree.
[Change]
New figure
“A QFlow scan on a plane perpendicular to blood flow was performed to obtain the cross-sectional images of the vascular structure of lower extremity at three levels (Figure 2). We analyzed the same setpoints for every patient.
Figure 2. QFlow scan was performed at three levels to obtain the cross-sectional images of the vascular structure of the lower extremity. Postprocessing analysis drew the region of interest on the vascular lumens at the setpoints of both external iliac veins, femoral veins, greater saphenous veins, and popliteal veins. Each setpoint of QFlow analysis can generate eight flow-related variables.
variables.
Quantitative analysis was performed by drawing the region of interest on the vascular lumens (covering the whole lumen) at each setpoints, including both external iliac veins, femoral veins, greater saphenous veins, and popliteal veins. QFlow variables included stroke volume (SV), forward flow volume (FFV), backward flow volume, regurgitant fraction, absolute stroke volume, mean flux (MF), stroke distance (SD), and mean velocity (MV) (Figure 3)”. [page 4 line 117-130]
[Comment 4]
You state that SD and MV can separate pts with occlusive from non-occlusive venous disease in EIV. Why only in EIV – do you have an explanation for this finding?
[Answer 4]
The possible explanation is that both the femoral vein lesions and EIV lesion could decrease the SD/MV value in the EIV. The femoral vein lesion may have less impact on the EIV QFlow. However, we did not found sufficient evidence to state that hypothesis.
[Change]
No change made.
[Comment 5]
You state that results are ratios between affected and unaffected leg. However, this implies that only patients with unilateral venous leg disease was enrolled? How can a ratio be above 1?
[Answer 5]
Other etiology like venous reflux and congenital anomaly, may paradoxically increase the QFlow parameter in the morbid limbs.
[Comment 6]
You should do Bonferroni-correction of data when so many tests are performed. This would most likely influence your final results.
[Answer 6]
We had performed the Bonferroni-correction and revised the results and the tables.
[Comment 7]
How were data collected? You did analyses off-line by one operator? Blinded? Automated? Inter- intraobserver variation?
[Answer 7]
All MRI analyses was performed by one operator (CWC). By drawing the region of interest (ROI) on the vascular lumens (covering the whole lumen), each ROI automatically generated flow-related variables by post-processing software in the computer at the workstation.
There is a possible disagreement between the observers (two sonographers, YHT and YKH) and between the imaging methods (ultrasound and TRANCE MRI), respectively. According to our prior study, inter-rater reliability between TRANCE MRI and doppler USG showed substantial agreement (Cohen kappa coefficient, 0.73). These differences between the observers may lead to grouping errors and affect the results of the study. Second, the lack of a reference method of hemody-namic measurement is a big weakness in this study.
[Change 7]
We had added the aforementioned description in the “Study Limitations”.
[Comment 8]
Table 2 is not very informative. Which are and which aren’t TRANCE findings?
[Answer 8]
All findings in Table 2 are diagnosed based on TRANCE MRI. According to this comment, we had deleted the original Table 2 and described the notable findings in this revised manuscript.
“TRANCE MRI revealed more cases of May-Thurner Syndrome (MTS) (18:5) and DVT (5:1) in the obstructive group comparing with the non-obstructive group. There are more varicose veins in the non-obstructive group (n=22) compared with the ob-structive group (n=14). Varicose veins in the non-obstructive group (17/22) had a trend accompanying prominent great saphenous veins compared with the obstructive group (1/14).” [page 6]
[Comment 9]
As stated above the results are difficult to read due to many abbreviations in Table 3. Table 3 should be better designed.
[Answer 9]
We have reduced the use of abbreviations in the table. For the simplicity of reading, we redesigned the original Table 3 and converted it to the new Table 2 and Table 3.
[Comment 10]
Don’t repeat results in the discussion. In discussion is missing the comparison to results of other groups.
[Answer 10]
This study has only two groups, and we simply the discussion in this version, removed some detail statistics already written in the result.
[Comment 11]
When abbreviations are introduced, then spelled out first time. Eg. missing for FV in abstract.
[Answer 11]
Thank you for this correction. We had checked it and revised the manuscript throughout the article.
Thank you for the informative and careful reviewing our manuscript. We learn lot during the revision this article. Please contact me at the following address for additional information.
Sincerely,
Yao-Kuang Huang, MD, PhD
Division of Thoracic and Cardiovascular Surgery
Chia-Yi Chang Gung Memorial Hospital, Putz, Taiwan.
Fax: 886-975368209
E-mail: huang137@icould.com

Round 2
Reviewer 1 Report
The Authors included necessary information in the text. At the present form, the study is nicely read. I do not have any additional comments.
Reviewer 2 Report
No further comments.